## Pathogens

# Stylet cuticular gene-directed mutagenesis impairs the pea aphid vector capacity to transmit a plant virus

Yu Fu[1], Maëlle Deshoux[1], Bastien Cayrol[1], Sophie Le Blaye[1], Emma Achard[1], Sylvie Hudaverdian[2], Romuald Cloteau[2], Elodie Pichon[1], Elian Strozyk[1], Nathalie Prunier-Leterme[2], Emmanuelle Jousselin[3], Nicolas Sauvion[1], Gaël Thébaud[1], Gaël Le Trionnaire[2], Stefano Colella[1]*, Marilyne Uzest[1]*

1 PHIM Plant Health Institute, University Montpellier, INRAE, IRD, CIRAD, Institut Agro, IRD, Montpellier, France, 2 IGEPP INRAE, Institut Agro, University of Rennes, Le Rheu, France, 3 CBGP, INRAE, CIRAD, Institut Agro, IRD, University Montpellier, Montpellier, France

* stefano.colella@inrae.fr (SC); marilyne.uzest@inrae.fr (MU)

## Abstract

Aphids are major agricultural pests, notably because they transmit nearly 30% of known plant viruses, including non-circulative ones. They can be collected and dispersed rapidly among crops while aphids feed on infected plants. Most of these viruses are retained on receptors located on the cuticle of the stylet tip. The acrostyle, a cuticular micro-territory at the apex of aphid stylets, has been identified for its ability to retain the cauliflower mosaic virus (CaMV). The acrostyle displays cuticular proteins, known as stylins, with exposed domains accessible at the virus-vector interface. RNAi-mediated silencing of Stylin-01 designated this protein as the prime candidate receptor of CaMV. However, the results were incomplete due to the transient effect and highlighted the need for stable mutants to advance our knowledge and validate these putative virus receptors. Here, we characterized the phenotype of two pea aphid Stylin-01 mutant lines, the first generated with CRISPR/Cas9 in this hemipteran. We showed that Stylin-01 mutations significantly disrupt CaMV transmission and impair the acrostyle's ability to bind the CaMV helper protein P2. Stylin-01 mutations also reshape the distribution of other stylins on the surface of mutant aphid stylets. In addition, Stylin-02, the putative ortholog of Stylin-01, is overexpressed in the mutant lines, pointing out a potential partial complementation of Stylin-01 in its structural role but not for virus transmission. In conclusion, this study, using the first stable aphid mutant lines, allows the characterization of the central role of Stylin-01 virus receptor in CaMV transmission.

**Data availability statement:** All relevant data are within the paper and its Supporting Information files.

**Funding:** This work was funded by the French National Research Agency (ANR-15-CE20-0011 to MU ; and ANR-21-CE20-0001 to MU), the Bill and Melinda Gates Foundation (GCEag, OPP1130147 to MU), the StylED project (défi clé RIVOC Occitanie Region, Université of Montpellier to EJ and SC). YF acknowledges the Chinese Scholarship Council (CSC Grant No. 202103250004) and the French National Institute Research for Agriculture, Food and Environment INRAE, for financial support of her PhD studies. The funders had no role in the study design, data collection and analysis, decision to publish, or preparation of the manuscript.

**Competing interests:** The authors have declared that no competing interests exist.

## Author summary

Aphids are severe agricultural pests, well known as major vectors of hundreds of plant viruses. They feed on plant sap using their specialized mouthparts, the stylets. At their tip they house the acrostyle which is a cuticular cell-free micro-structure with adhesive properties featuring the stylin proteins that bind aphid effectors, critical molecules for plant-insect interactions. Non-circulative viruses have evolved to bind aphids' stylet tips and stylins to be transported from plant to plant. Understanding stylins' role in virus-vector-plant interactions and identifying virus receptors constitute a stepping stone for developing innovative crop protection approaches. Using the first CRISPR-Cas9-generated aphid mutants, we demonstrate the crucial role of Stylin-01 as a receptor of the plant virus CaMV.

## Introduction

Aphids are hemipteran phloem-feeders that are major pests for agriculture. They are found in most cultivated ecosystems, mainly in temperate regions, but not exclusively [1,2]. In addition to the direct damages inflicted on the plants on which they pullulate, they are vectors of a wide variety of plant pathogenic viruses, significantly impacting plant health and crop yields [3,4]. The transmission mechanisms have been extensively studied in aphids as they exhibit the broadest diversity of virus-vector interaction categories among other insects [5]. Aphids are particularly effective vectors of non-circulative viruses comprising hundreds of viral species. These viruses bind on the cuticle of the aphid's mouthparts either in the foregut or more commonly in the maxillary stylets where they are specifically and transiently retained [6,7]. Unlike circulative viruses, non-circulative ones need not be internalized in their vector for transmission to new plants. Aphids can acquire and inoculate non-circulative viruses within seconds during feeding probes they make to test the suitability of both host and nonhost plants before eventually fleeing to more compatible hosts [8–10]. Controlling viral epidemics in these conditions is challenging. Developing approaches that effectively prevent or inhibit virus transmission by aphids requires an in-depth understanding of virus-vector interactions, including the identification of virus receptors in vectors' mouthparts.

Cauliflower mosaic virus (CaMV) is the virus model for which work on receptor identification is the most advanced. The CaMV is transmitted by aphids using a helper strategy. Its retention in the insect stylets is mediated by a helper component, the non-structural viral protein P2 [11]. Its aphid receptors are located on the surface of the acrostyle, a cuticular micro-territory at the apex of aphid stylets with unique surface properties distinct from the rest of the cuticle [12,13]. This area also likely harbors receptors for numerous other non-circulative viruses [8]. Six cuticular proteins, named stylins, were identified at the surface of the acrostyle [14,15]. Therefore, they are all candidate receptors of plant viruses. Five of them (Stylin-01, 02, 03, 04, and 04bis) belong to the most prominent cuticular protein family defined by the presence of the conserved Rebers and Riddiford Consensus motif (R&R) [16], known as CPR family. These five stylins contain one of the three known distinct forms of the consensus

called RR-1 chitin-binding domain. CPR proteins particularly RR-1 types have recently emerged as key players in the vector transmission of circulative and non-circulative plant viruses [14]. Stylins at the acrostyle surface also interact with salivary effectors which suggests their original role in aphid-plant interactions [17] later hijacked by viruses to ensure their transport from plant to plant.

Stylin-01 and Stylin-02 were the first acrostyle proteins identified [14]. They share a high degree of sequence similarity, possess an almost identical C-terminal domain and are detected all over the acrostyle surface where the CaMV helper protein P2 binds [14]. The CaMV receptors are not exposed on the surface of a cell, but are embedded into an assembly of chitin nanofibres and specific structural proteins making up the cuticle of aphid cell-free stylets. Since obtaining folded functional stylins outside their natural environment is not feasible, classic protein-protein interaction approaches to determine which stylin interacts with the CaMV P2 protein are not achievable [14]. The role of Stylin-01 and Stylin-02 in CaMV transmission has been investigated in the green peach aphid *Myzus persicae* using RNAi-mediated knockdown. Although silencing of Stylin-01 was only partial, transient and variable, this approach confirmed this protein as a prime candidate for CaMV receptor as virus transmission by aphids was reduced following siRNA treatment. The same was not true when silencing Stylin-02 [14]. RNAi-mediated gene silencing does not allow formal demonstration of gene function. To overcome this limitation, we took advantage of the CRISPR-Cas9 mutagenesis approach to develop Stylin-01 mutant lines in the pea aphid *Acyrthosiphon pisum*, a vector aphid species able to transmit CaMV [18].

Here, we present a comprehensive characterization of the phenotypic traits of two Stylin-01 edited aphid lines, a complete knock-out and a mutant that produces a protein modified at its exposed C-terminal domain. Using these unique mutant lines, we demonstrate that a fully functional Stylin-01 is needed for efficient CaMV transmission by aphids. Our work significantly advances the understanding of the role of this cuticular component of the acrostyle, central organ in plant-virus-vector interaction.

## Results

### Characteristics of the Stylin-01 mutant lines used in this study

To investigate the role of the RR-1 cuticular protein Stylin-01 in CaMV transmission and determine whether disrupting *stylin-01* gene would lead to a total absence of virus transmission, we used stably edited Stylin-01 mutant *A. pisum* lines generated in a previous study [18]. We selected three genotypes from the same parental crossing for our experiments: the wild-type (WT) unmodified sister line used as a control and two mutant lines, Sty01-KO and Sty01-Cter (respectively L2 and L14 lines in Le Trionnaire et al. 2019) [18]. In the Sty01-KO line, a deletion in the first allele and multiple re-arrangements in the second one generated frameshift mutations, changes in the amino acids (AA) sequences, and early translational stop signals. This resulted in truncated alleles and total loss of function (Figs 1A and S1). In the Sty01-Cter line, allele 1 encodes a shorter protein lacking the 11 C-terminal AA that form part of the acrostyle surface-exposed domain susceptible to interacting with viruses. Allele 2 encodes a protein in which the last 11 AA have been replaced by an entirely new sequence of 12 AA not present in the wild-type protein (Figs 1A and S1). The 3D structures of the Sty01-Cter mutated Stylin-01 proteins predicted by AlphaFold2 closely resemble the one of Stylin-01 wild-type, with a C-terminal domain partly folded in the alpha helix (in red, Fig 1A). The Sty01-KO mutant aphids no longer produce Stylin-01 protein and synthesize a stylet bundle devoid of this RR-1 cuticular protein (Figs 1B, S2 and S1 Table). In contrast, the mutated Stylin-01 protein is produced in Sty01-Cter aphids. The isoform encoded by allele 2 is detected in whole-body crude extracts (Fig 1B) and is detected at the apex of maxillary stylets by immunofluorescence microscopy (Fig 1C). The three aphid lines have a similar developmental time (for details, see S3 Fig and S2 Table).

### Disruption of Stylin-01 impairs the capacity of aphids to transmit the cauliflower mosaic virus

We investigated the impact of Stylin-01 mutations on CaMV transmission efficiency by conducting transmission test experiments with cohorts of wild-type and Stylin-01 mutant aphids. Our results indicate that the transmission of CaMV was

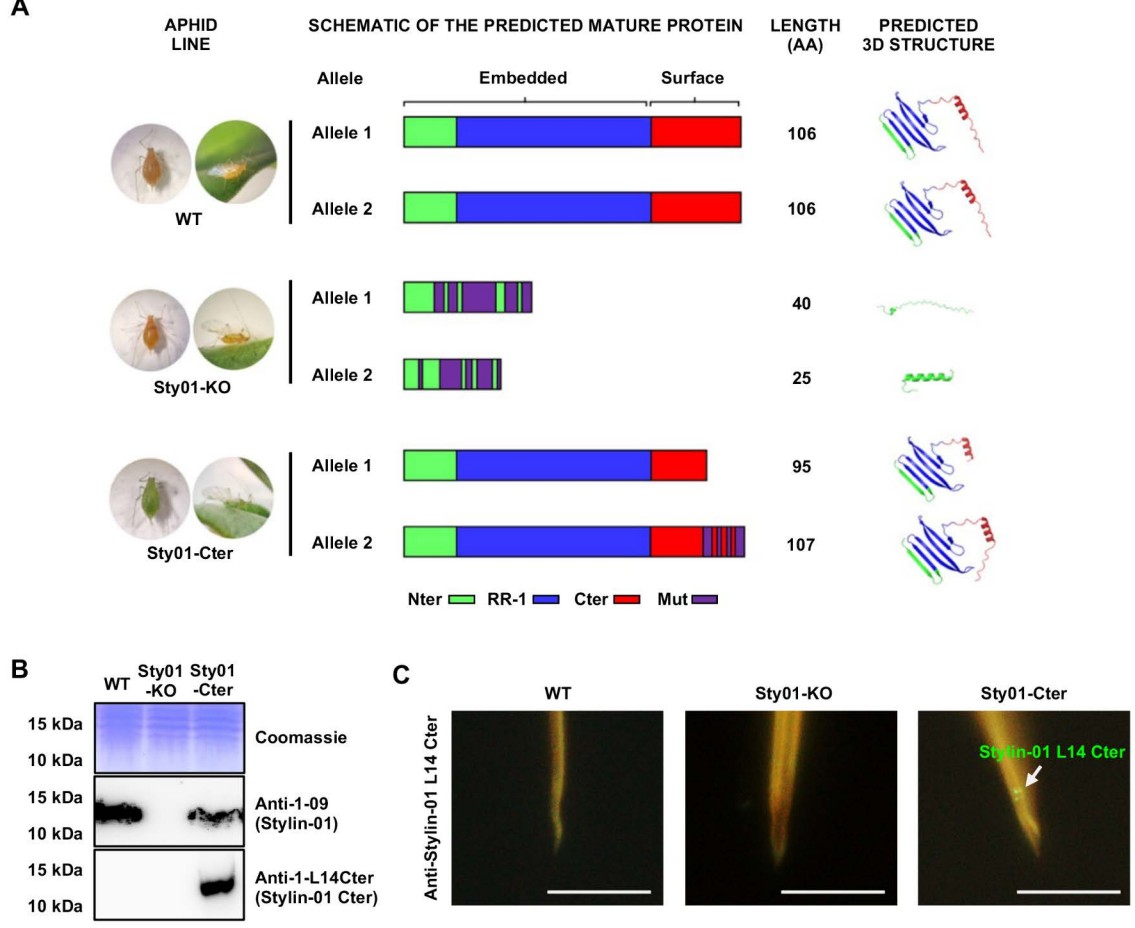

**Fig 1. CRISPR-Cas 9 stylin-01 edited aphid lines (Sty01-KO and Sty01-Cter) and wild-type non edited line (WT). (A)** Schematic representation of the domains of the mature proteins and their respective 3D structure predicted by AlphaFold2. The N-terminal (Nter), C-terminal (Cter) and RR-1 chitin-binding domains are indicated in green, red and blue respectively. Mutations in the amino acid sequences (Mut) are indicated in purple. Credit images Yu Fu. **(B)** Detection of Stylin-01 proteins in *A. pisum* lines by western blot analysis. Whole-body crude extract proteins from WT, Sty01-KO, and Sty01-Cter aphids (lanes 1, 2 and 3, respectively) were analyzed by SDS-PAGE and stained with Coomassie blue to compare the protein amount loaded on the gel, or transferred onto a nitrocellulose membrane prior to immunolabeling with antibodies anti-1-09 or anti-1-L14Cter antibodies (S2A and B, S1 Table). **(C)** Detection of Stylin-01 Cter mutant protein in dissected stylets of the 3 aphid lines using immunofluorescence labeling and confocal micros-copy Alexafluor-488 conjugated secondary antibodies were used to label anti-1-L14Cter antibody. The mutant protein is detected as green dots at the tip of maxillary stylets of Sty01-Cter aphids (indicated with a white arrow). Stylets of the wild-type or Sty01-KO aphids are not labeled by this antibody. Bars: 10 μm.

not completely abolished, regardless of the mutations considered. However, both mutants were impaired in their capacity to transmit CaMV compared to control aphids (Fig 2). The lowest transmission efficiency was obtained for the Sty01-KO mutant with a mean of 3.8±0.8% transmission efficiency. This result shows a highly significant 84% reduction in transmission efficiency compared to the wild-type TNJ4 (23.6±1.7%, P<0.001). Sty01-Cter aphids also showed an intermediate transmission efficiency lower than WT (13.5±2.5% transmission, P=0.011) but higher than Sty01-KO (P=0.001).

## Mutations compromise the acrostyle's ability to bind the CaMV P2 helper protein

We tested whether the decrease in transmission efficiency reflected a reduced ability of the mutated aphid stylets and acrostyle to interact with the CaMV P2 helper protein. We therefore incubated dissected stylets of WT, Sty01-KO, and

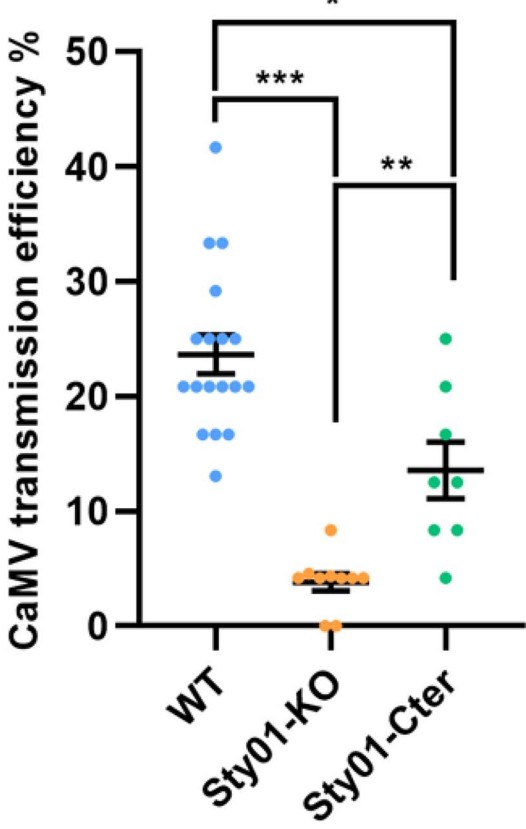

**Fig 2. Impact of Stylin-01 mutations on CaMV transmission efficiency.** The values are presented as the means ± SE of all independent replicates. Asterisks indicate significant differences according to binomial generalized linear model followed by a post-hoc Tukey's honestly significant difference (HSD) test (GLM, *P < 0.05; **P < 0.01; ***P < 0.001).

Sty01-Cter aphids with the viral protein fused to GFP (P2-GFP). Maxillary stylets were grouped into 3 classes (positive, partial, negative) according to the intensity and location of the observed labeling (Fig 3A, see Materials and Methods for details). The results show a significant reduction in the number of positively labeled stylets for both mutant lines compared to WT aphids (Fig 3B). In contrast with the maxillary stylets in WT that were all able to bind P2-GFP strongly along the acrostyle surface, 23.3% and 19.8% of the Sty01-KO and Sty01-Cter mutants' stylets respectively showed a statistically significant P2 binding defect. Consistent with the virus transmission defect, our results indicate that the stylets of mutant aphids have altered binding capacities for CaMV P2.

**No ultrastructural changes in the acrostyle of mutant aphids**

To examine whether the introduced mutations affected the ultrastructure of aphid stylets, particularly the apical part of maxillary stylets housing the acrostyle, we conducted a series of observations of the stylet tip using transmission electron microscopy (TEM). The TEM approach allows the observation of the acrostyle as a thin, distinguishable electron-dense area in the common food/salivary canal of *A. pisum* and other aphid species on stylet tip cross-sections [12,13]. As shown in the images (Fig 4), the maxillary stylets of both mutants and WT exhibit a clear layer surrounding the stylet cuticle forming a darker line in the bottom bed of the stylet typical of the acrostyle. Therefore, TEM could not highlight any detectable difference in this area at this scale.

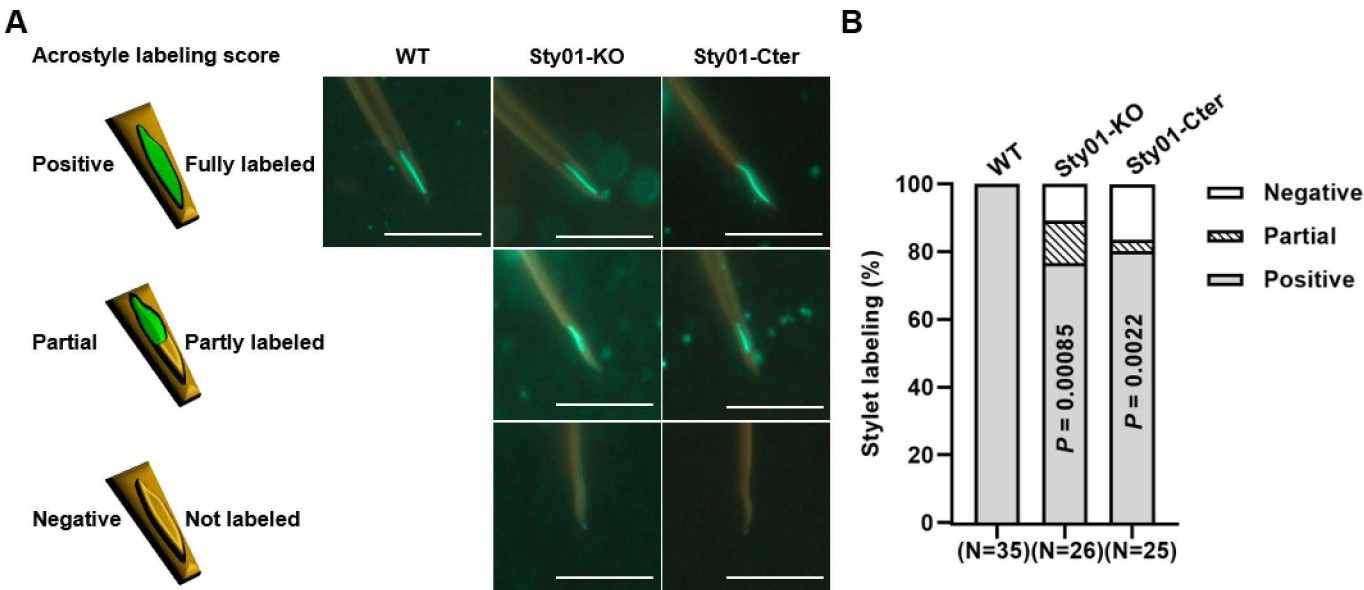

**Fig 3.** *In vitro* interaction assays between dissected aphid stylets and the viral protein CaMV P2 fused to Green Fluorescent Protein (P2-GFP). **(A)** Representative images of the detection of P2-GFP on the acrostyle of dissected stylets of WT, Sty01-KO and Sty01-Cter adults. Stylets were scored (positive, partial, negative) according to the detection and localization of the green fluorescence labeling as shown in the schematic diagram on the left panel. Scale bars represent 10 µm. **(B)** Proportion of maxillary stylets labeled for each category out of the total number of maxillary stylets observed. The number of positive stylets was significantly lower in the two mutant lines Sty01-KO and Sty01-Cter than in the wild-type aphid line (P-values from a binomial GLM), "N" indicates the total number of maxillary stylets observed for each treatment from three (Sty01-KO, Sty01-Cter) or four (WT) biological replicates.

### Modification of stylins' distribution at the surface of stylets of mutant aphids

Using a set of well-defined specific antibodies (S2 Fig and S1 Table), we further investigated the distribution of RR-1 stylins (Stylin-01, 02, 03, 04, and 04bis) in the maxillary stylets of the wild-type and mutant lines using immunofluorescence microscopy. The anti-1-11 antibody directed against the 15 C-terminal AAs of Stylin-01 and Stylin-02 can detect both proteins (S2A and 2C Figs). Our results show that, in both mutant aphid lines, stylet labeling with anti-1-11 antibody is similar to that observed in wild-type aphids (Fig 5), indicating that in the absence of Stylin-01, the C-terminal domain of Stylin-02 is highly accessible all along the acrostyle surface of their stylets. The anti-1-15 antibody targeting Stylin-03 protein reveals that the distribution differs in the stylets of the three aphid lines. Stylin-03 protein is mainly detected as a labeled broken line at the surface of the acrostyle in 74% of the WT and 91% of the Sty01-Cter stylets. On the other hand, in the Sty01-KO mutant Stylin-03 is weakly detected and located at the edges of the apical part of maxillary stylets (Fig 5, S3 Table).

The N-terminal domain of Stylin-04 and 04bis is similarly detected at the apex of maxillary stylets of wild-type and mutants, but with a weaker signal on the acrostyle edge and in its upper part in the vicinity of the food canal (Fig 5).

### Loss of Stylin-01 is associated with the upregulation of Stylin-02 during stylet synthesis

To evaluate whether the introduced mutations affected the expression of other RR-1 Stylins, we quantified their expression during adult stylet synthesis based on our stylet development analysis (i.e., towards the end of the last nymphal stage N4) [19]. As the developmental time of the aphid nymphs was almost identical for all the lines (S2 Table and S3 Fig), we sampled all N4 nymphs at the peak of Stylin expression 46 hours post-molt as precisely determined in a previous study [20]. Thus, synchronized WT, Sty01-KO, and Sty01-Cter aphids were followed individually from the end of N3

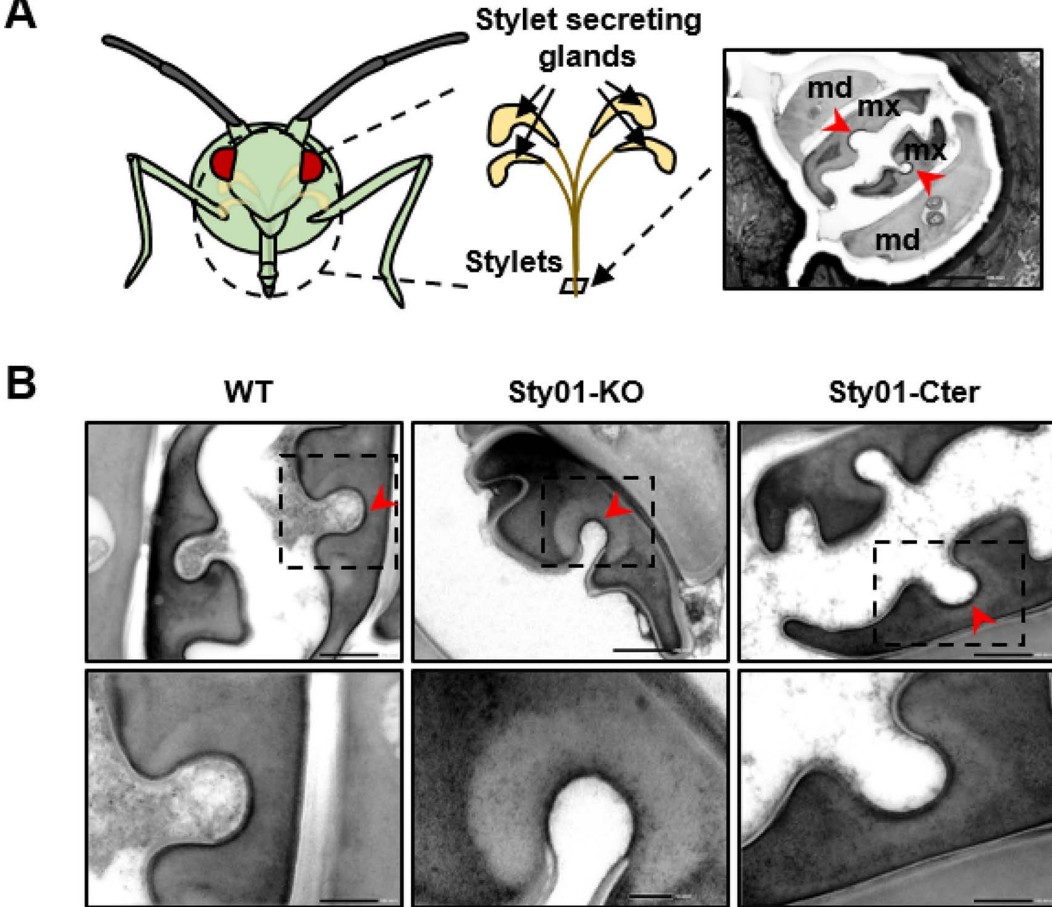

**Fig 4. Observation of the stylet ultrastructure in wild-type and Stylin-01 mutant aphids using transmission electron microscopy (TEM). (A)** Schematic representation of the approximative position of the cross sections of the apical part of aphid stylets observed under TEM in **(B)**. The stylet bundle visible on the electron micrograph shows the two mandibular stylets (md) surrounding the two maxillary stylets (mx). **(B)** Electron micrographs of cross sections of the maxillary stylets of each aphid line (upper panel), observed at higher magnification in the bottom panel. The acrostyle, visible as an electron-dense area in the maxillary stylets, is indicated with a red arrowhead in (A) and (B, upper panel). Bars: 500 nm in **(A)**, 200 nm in (B, upper panel), 50-100 nm in (B, bottom panel).

stage to the time of sampling. We performed RT-qPCR analyses using specific primers for *stylin-01*, *-02*, and *-03*, and primers amplifying indiscriminately the highly homologous *stylin-04* and *stylin-04bis* transcripts (S4 Table). The expression patterns of *rr1 stylins* genes in WT and Sty01-Cter lines were similar, with *stylin-01* being the most highly expressed. A residual expression of *stylin-01* gene is detected in the Sty01-KO line as expected, as the PCR primers for *stylin-01* target a region upstream of the mutations in the two alleles. On the other hand, Stylin-02 was the most highly expressed stylin in the Sty01-KO mutant (Fig 6). Consistently, more Stylin-02 protein was detected in whole-body crude extracts of mutant aphids than wild-type aphids (S4 Fig). In contrast, the expression of *stylin-03, stylin-04* and *04bis* transcripts appeared to be unaffected in Sty01-KO and Sty01-Cter mutants (Fig 6). Our results indicate that *stylin-02* expression is upregulated in Sty01-KO nymphs N4 during adult stylet synthesis.

### Disruption of Stylin-01 has no impact on the feeding behavior related to CaMV acquisition and inoculation

To determine if changes in the acrostyle protein surface of pea aphid mutants affect their feeding behavior during CaMV acquisition and inoculation, and subsequently virus transmission, we monitored the feeding behavior of cohorts of both

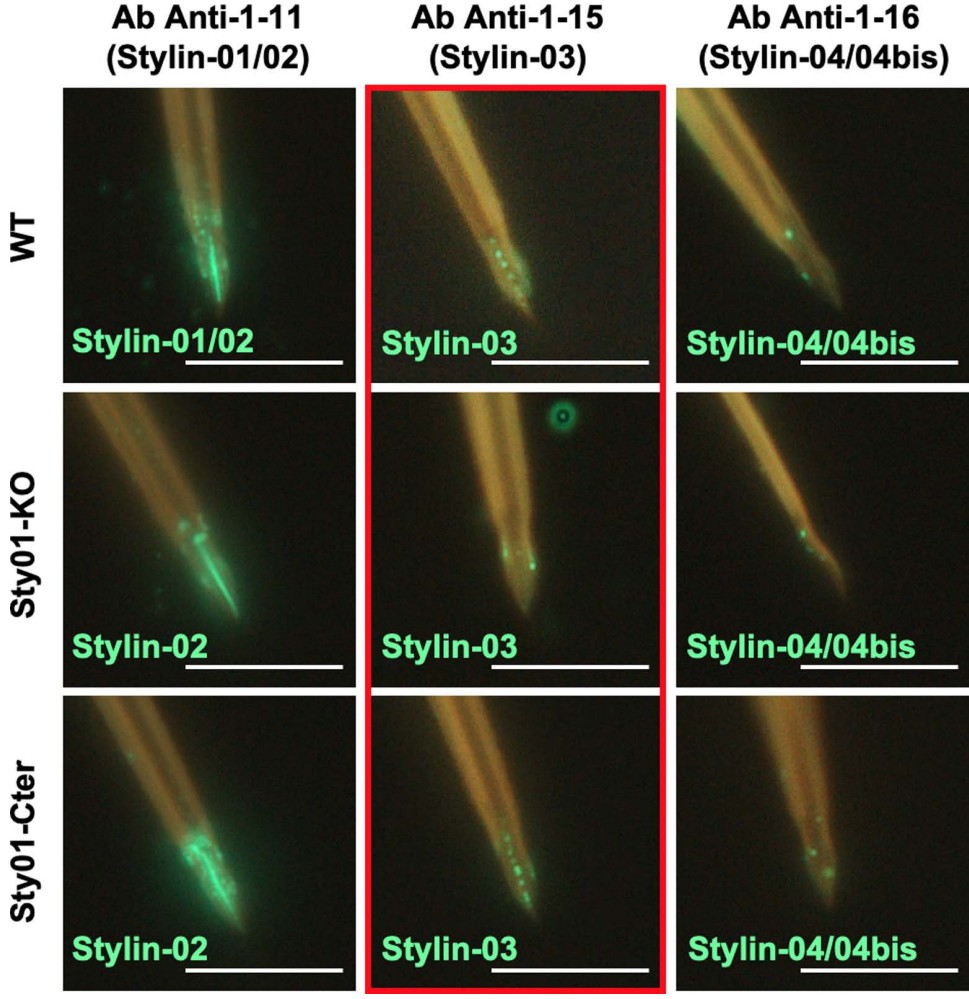

**Fig 5. Distribution of RR-1 stylins in wild-type and Stylin-01 mutant aphids.** Representative images of labeling of stylin peptides observed in maxillary stylets of WT, Sty01-KO and Sty01-Cter aphids. The primary antibodies used (Ab anti-1-11, Ab anti-1-15, Ab anti-1-16) and the stylins they theoretically target (Stylin-01, -02, -03, -04/-04bis) is indicated above the images (S2 Fig and S1 Table). The stylins detected in aphid stylets are indicated in green at the bottom of each image. The main difference observed at the acrostyle surface is outlined in red. Bars: 10 μm.

wild-type and mutant nymphs by electropenetrography (EPG). CaMV can be acquired during brief (i.e., a few seconds) intracellular punctures in the epiderm or mesophyll of infected plants [21] and inoculated within seconds when the aphid feeds on another plant [22]. We therefore used short recording times compatible with non-circulative transmission [23].

We first compared more precisely the variables related to CaMV acquisition: the time for stylets from the start of the first probe (waveform C) to reach the first cell puncture in the mesophyll (waveform pd) (t>1pd) [24], and the duration of subphase II-3 of the first intracellular puncture in mesophyll tissue (d_1pdII-3) corresponding to ingestion of the cell contents and virus uptake [8] (Fig 7A). We observed that most of the nymphs had achieved a first intracellular puncture within 20 seconds after the start of the first probe, whatever the aphid line. The duration of the feeding behavior sub-phase II-3 showed a similar distribution pattern in all individuals of the three *A. pisum* lines, (Fig 7B and S5 Table).

We then focused on EPG variables related to virus inoculation into young turnip plants (Fig 7C). We observed no significant difference in feeding behavior among the three aphid lines during the intercellular path to the first punctured cell and

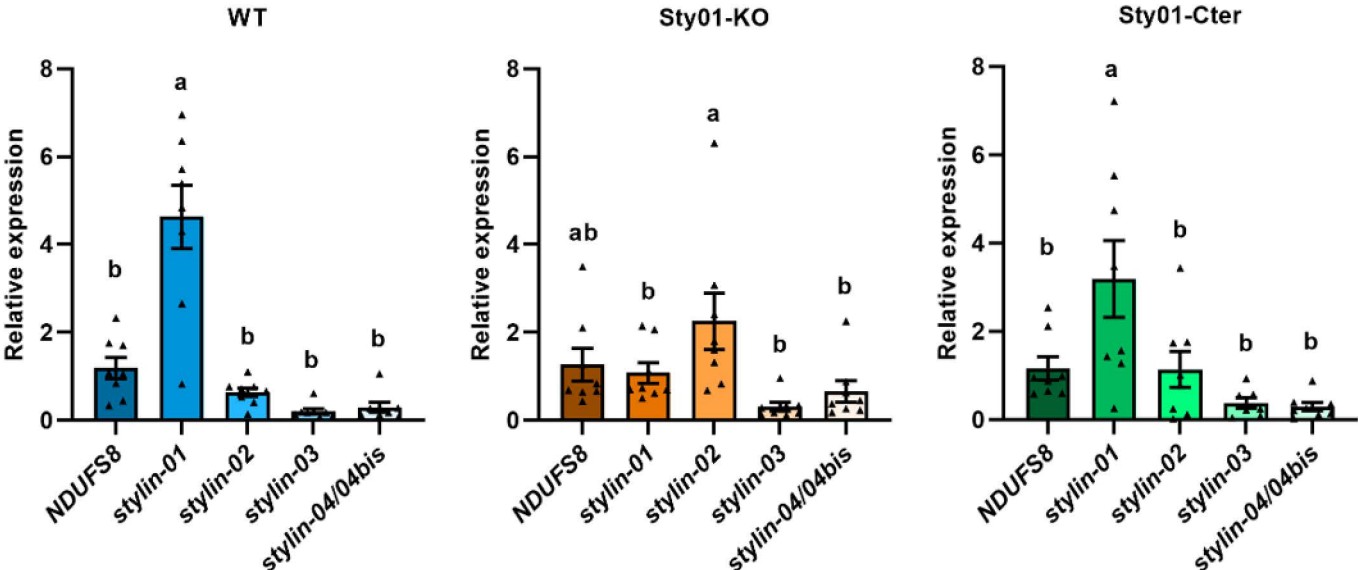

**Fig 6. _Stylin-02_ is upregulated in the Sty01-KO mutant during the synthesis of adult stylets.** Relative expression levels of stylins at N4 instar 46 h post-molt were measured in WT and mutant lines. The gene encoding NADH dehydrogenase [ubiquinone] iron-sulfur protein 8 (NDUFS88) was used as internal control to compare the expression of all RR-1 stylins in each aphid line. The values are presented as the means ± SE (n = 7-8 biological replicates from two independent experiments, different letters indicate significant differences according to one-way ANOVA followed by the least significant difference (LSD) test.

in the time spent in this first cell. In particular, the duration of subphase II-2, previously shown to be associated with CaMV inoculation into plant cells [9] were similar (Fig 7D and S5 Table).

### Phylogenetic analyses of RR-1 stylins

Alignment of Stylin-01 (NP_001155786.1) and Stylin-02 (NP_001156143.1) pea aphid proteins showed a high degree of similarity [14]. In 11 Aphididae species available in GenBank, we obtained putative ortholog sequences for the five RR-1 known stylin genes of the pea aphid [15]. Using Orthofinder [25], these sequences were assigned to three ortho-groups, one containing Stylin-01 and 02, one for Stylin-03, and one for Stylin-04 and 04bis. Phylogenetic analyses of the Stylin-01 and 02 othogroup separated the assigned proteins into two branches, one for each stylin (S6 Table and S5A Fig). In each branch, the protein tree more or less parallels the more recently established Aphididae species phylogenetic history (S5B Fig) [26,27]. The only incongruence is the relative position of the two species outside the Aphidinae subfamily (i.e., _Sipha flava_ and _Cinara cedri_), which varies in each subtree.

### Discussion

RR-1 cuticular proteins are major components of the acrostyle surface at the apex of aphid stylets, playing a crucial functional role [15]. In this cell-free cuticular micro-territory RR-1 cuticular proteins enable transient binding of endogenous and exogenous molecules such as salivary effectors and viruses. Non-circulative viruses have evolved to interact specifically with the motifs exposed at the surface of the acrostyle, thereby ensuring their transport between plants [14,17]. Receptors of viruses are prime targets for innovative control methods, but their identification in insect vectors remains a considerable technical challenge to overcome, especially for those embedded in the cuticle of the mouthparts. In the absence of standard experimental approaches to characterize virus-receptor interactions, including both _in vitro_ and _in vivo_ heterologous

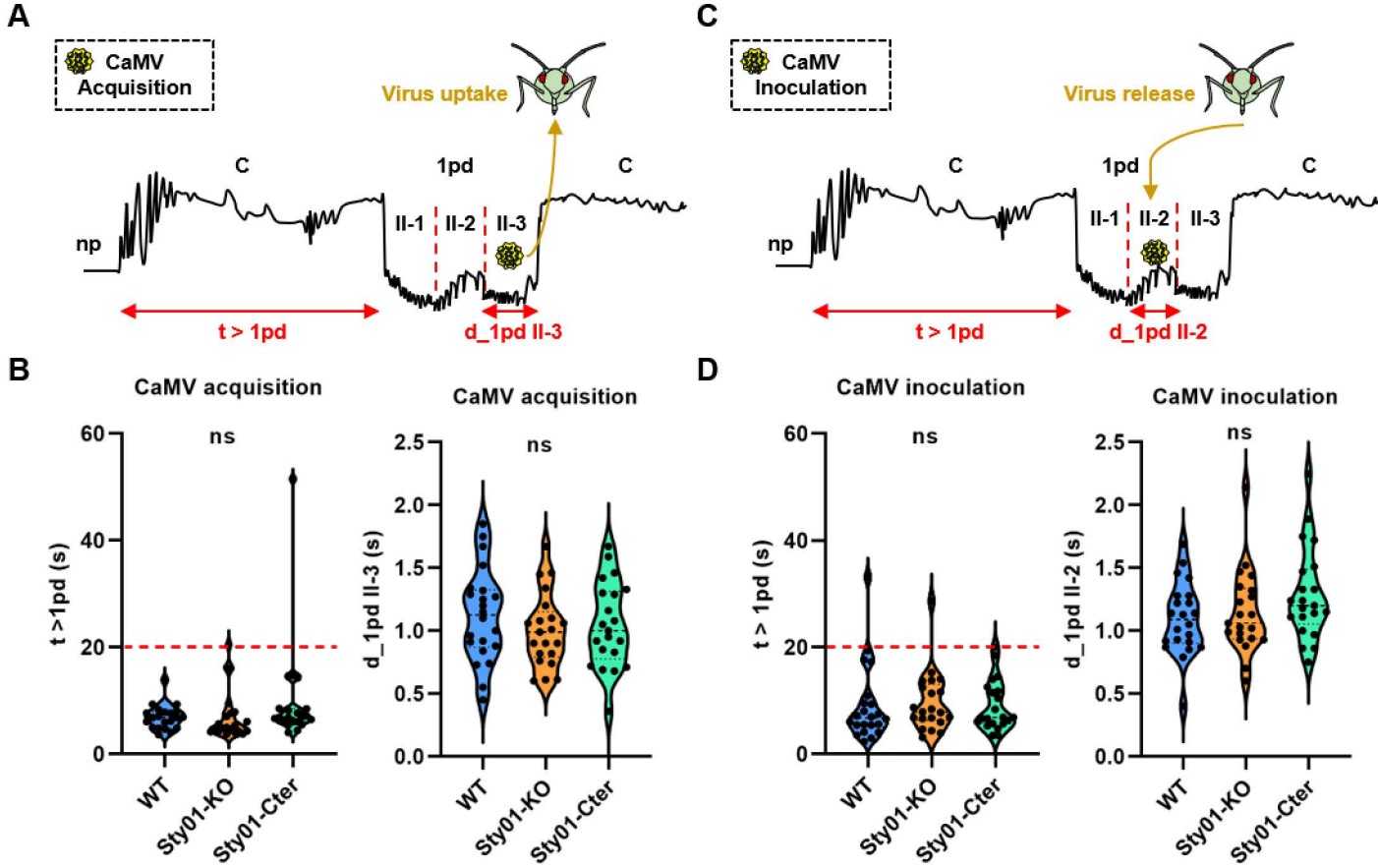

**Fig 7. Feeding behavior of wild-type (WT) and mutant *A. pisum* lines related to CaMV acquisition and inoculation. (A, C)** Schematic diagrams illustrating the waveforms related to CaMV acquisition and inoculation, respectively. The duration of first C to first pd (time from the beginning of the first probe to the first intracellular puncture) during CaMV acquisition (B) and inoculation (D) show no difference between all aphid lines (n = 21-22, Kruskal-Wallis test, ns = not significant at the 5% level). Below the dotted line are most of the aphids that completed the sequence from the first C to the first pd. The duration of the sub-phase II-3 in the first pd related to ingestion of cell content (B) or the duration of subphase II-2 in the first pd related to CaMV inoculation (D) are similar in all aphid lines (n = 21-22, Kruskal-Wallis test, ns = not significant at the 5% level).

systems, the role of some RR-1 candidate receptors in virus transmission could until now, be assessed only by RNAi mediated gene expression silencing [14].

In the present study, we used CRISPR-Cas9 edited Stylin-01 stable mutant lines in *A. pisum* [18] to advance our knowledge of the role of Stylin-01 as a receptor of CaMV. The mutations introduced in the *A. pisum* genome had no apparent impact on the mutant aphids' developmental time and viability, as colonies could be maintained clonally for years under laboratory conditions. The availability of the complete knock-out and a mutant for the C-terminal exposed domain of Stylin-01 allowed us to study its role in plant virus transmission. As expected, the Sty01-KO line exhibits a stronger phenotype than the Sty01-Cter mutant, thus allowing a more comprehensive characterization of the Stylin-01 function.

The ability to transmit the CaMV is almost completely lost in the Sty01-KO mutant line, and highly reduced in the Sty01-Cter compared to the wild-type (Fig 2). These findings are consistent with Webster *et al.* [14], who demonstrated that decreasing significantly *stylin-01* expression in the green peach aphid *Myzus persicae* via RNAi led to a reduction in CaMV transmission. The intermediate phenotype of the Sty01-Cter mutant in CaMV transmission indicates a potential

role for the C-terminal helix in the binding of the virus to the acrostyle. Only the last 11 C-terminal amino acids from the extended 33 AA C-terminal domain in the Sty01-Cter mutant are missing in allele 1 and are different in allele 2 compared to the wild type. Based on the 3D AlphaFold prediction in both alleles, the exposed helix structure is conserved (Fig 1). Whether the helix itself or the whole domain is important for P2 binding would need further investigation. But unfortunately, interactions between the viral protein P2 and cuticular protein derived-peptides cannot be tested with classical approaches [14,28].

The observed transmission defect is not due to a change in the insect feeding behavior, as wild-type and mutant aphids could perform intracellular punctures within 20 seconds after initiating a probe in CaMV-infected and healthy plants, indicating their ability to potentially acquire and inoculate the virus.

We show here that the viral protein P2 can still bind to the maxillary stylets of both mutant aphids even if with less efficiency, indicating that Stylin-01 is not the sole acrostyle protein with CaMV-binding activities. Disrupting *stylin-01* does not change the overall stylet ultrastructure but affects the surface distribution of other stylins. The localization of Stylin-02 on the surface of the Sty01-KO mutant stylets is compatible with the localization of CaMV P2 binding sites, while that of Stylin-03 or Stylin-04/04bis is not (Fig 5). The high similarity between Stylin-01 and -02 in their exposed domains prevents the development of specific antibodies. The antibody recognizing both putative orthologs showed no difference in acrostyle labeling between mutants and wild-type lines (Fig 5). Differences in the labeling of mutant lines could have been expected, particularly in the Sty01-KO one where only Stylin-02 is present. The fully labeled acrostyle in the KO mutant suggests that Stylin-02 may compensate for the missing Stylin-01 protein in organizing this cuticular region that is essential for plant-aphid interactions. This potential role is further supported by Stylin-02 overexpression during stylet biogenesis in the mutant lines and by evolutionary analyses that indicate that Stylin-01 and Stylin-02 are orthologs. Nonetheless, Stylin-02 distribution, even if aligned with P2 binding sites does not seem sufficient to complement the lack of Stylin-01 restoring CaMV transmission in the KO mutant.

This study represents the first functional characterization of the role of a gene in a key life history trait of aphids using genome editing. Our work on CRISPR-Cas9 Stylin-01 mutants shows the predominant role of this acrostyle RR-1 cuticular protein in CaMV transmission by the pea aphid. At the same time, the observed low residual CaMV transmission in the Sty01-KO line shows that Stylin-01 is not the exclusive receptor for this virus. The role of stylins in virus transmission calls for further studies that would benefit from stable mutant lines that enable detailed phenotype characterization and gene function validation.

## Materials and methods

### Aphid lines, virus strain, and plants

The three sister lines of *Acyrthosiphon pisum* used in this study, the wild type line (WT) and the two *stylin-01* mutant lines Sty01-KO (full gene knock-out) and Sty01-Cter (Cter deletion) were generated in a previous study [18]. Aphids were reared in controlled conditions at 18 °C and a photoperiod of 16/8 h (day/night) on faba bean (*Vicia faba* cv. "Sutton"). The *cauliflower mosaic virus,* CaMV (isolate Cabb B-JI), was mechanically passaged on turnips (*Brassica rapa* cv. "Just Right"). Healthy and infected plants were maintained at 23/18 °C and a photoperiod of 16/8 h (day/night).

### Aphid lines developmental time assessment

To monitor the duration of nymph development, apterous adult aphids were placed on two-week-old faba beans. Experiments were conducted at 18 °C with a photoperiod of 16/8 h (day/night). Adults were allowed to produce one progeny, then eliminated. For each lineage (WT, Sty01-KO, Sty01-Cter), at least 10 nymphs were individually surveyed; the experiment was repeated twice. The insects were observed every 6 hours, and the duration of each instar was recorded as soon as the nymphs molted (S2 Table).

## Protein extraction and western blot analysis

For each *A. pisum* line, whole-body proteins were extracted from 10 adults in 80 µL phosphate-buffered saline [4.3 mM $Na_2HPO_4$, 1.4 mM $KH_2PO_4$, 137 mM NaCl, 2.7 mM KCl, pH 7.3]. Samples were separated under denaturing conditions in 15% SDS-PAGE. Gels were stained with Coomassie blue or were transferred onto nitrocellulose membranes (Amersham, Marlborough, USA) using Semi-Dry Blotter. After blocking in Tris-buffered saline (50 mM Tris and 200 mM NaCl, pH 7.4) supplemented with 0.05% Tween 20 (TBST buffer) and 5% (w/v) skimmed milk, membranes were incubated overnight at 4 °C with with anti-stylin primary antibodies diluted 1:1000. Membranes were finally incubated for 1 h at room temperature with goat-anti-rabbit IgG-HRP secondary antibodies (sc-2030; Santa Cruz Biotechnology, Dallas, TX) diluted 1:1500, blots were visualized with the G-BOX (Syngene, USA). Interactions were revealed using chemiluminescence according to the manufacturer's instructions (ThermoFisher Scientific).

## Aphid transmission assays

Two-day-old aphid nymphs (N2) were collected in glass tubes for 1 h starvation. They were allowed a 2-minute acquisition access period (AAP) on leaf L8 of a CaMV-infected turnip used as a virus source plant 21 days post-inoculation. The nymphs were immediately collected in a glass Petri dish and transferred to 8-day-old turnip test plants for a 3 h inoculation access period (IAP) followed by insecticide treatment. The following design was used in all experiments: 2 nymphs/test plant; 24 test plants/source plant. The aphid transmission efficiency was calculated by recording 28 days post-transmission assay the number of plants presenting viral symptoms relative to the total number of test plants used. At least six independent biological replicates were performed for all aphid lines.

## *In vitro* interaction assays on dissected stylets

For immunolabeling, dissected adult stylets were immunolabeled according to Webster and colleagues using primary antibodies at a 1:200 dilution (S1 Table) and secondary Alexa fluor-conjugated antibodies at a 1:400 dilution [28]. Stylets were mounted in Mowiol mounting medium on a glass slide. Images were taken with a confocal laser scanning microscope (ZEISS LSM900). For *in vitro* binding assays with the viral protein P2 fused to GFP (P2-GFP) produced in Sf9 cells, stylets from adult aphids were dissected and incubated with the protein as previously described [12]. Maxillary stylets were scored according to the intensity and location of labeling. They were considered "Positive" if they were strongly labeled on the entire surface of the acrostyle, "Partial" if the labeling was only present on part of the acrostyle, or "Negative" if no labeling was detected. The images were noted by two independent observers. For both types of interaction assays, batches of 8–16 maxillary stylets were observed for each aphid line and experimental condition. Three independent biological replicates were performed.

## Sample preparation for transmission electron microscope (TEM)

*Acyrthosiphon pisum* heads were severed from cold-anesthetized adults and fixed in fixative buffer (FB: 0.1 M cacodylate buffer, pH 7.2) containing 4% glutaraldehyde overnight at 4 °C. After rinses in FB followed by post-fixation in FB containing 1% osmic acid for 2h at 4 °C, samples were dehydrated in a graded acetone series. Finally, they were embedded in Epoxy resin (EmBed 812) using an automated microwave tissue processor for electron microscopy, Leica EM AMW. Sections 60 nm thick were observed in a JEOL JEM 1400 microscope.

## Reverse-transcription quantitative PCR

*Stylin* expression levels were measured in each aphid line at the pre-imaginal stage during the synthesis of adult stylets. N4 nymphs were collected at 46 h post molting, the exact time corresponding to the peak of expression of *stylin-01* and *stylin-02* genes in aphid heads [20]. To that end, wingless adults were put on young faba bean plants and allowed

to produce nymphs for 2 h. Cohorts of synchronized aphids were individually followed. N3 nymphs were isolated and screened regularly for N4 emergence. Nymphs were collected 46 h after the beginning of the N4 stage, and their head were immediately dissected. Pools of three heads without antennas were placed in an RNAlater solution (ThermoFisher Scientific, Waltham, MA, USA) and flash-frozen in liquid nitrogen. Samples were kept at -80 °C until use. Four pools of three heads were collected for each aphid line, and the whole process was repeated twice, resulting in a total of 8 pools of 3 aphid heads per line.

Total RNA was extracted using a RNeasy mini kit (Qiagen, Hilden Germany) and treated with RQ1 RNAse-free DNAse I (Promega Corporation, Madison, WI, USA). First-strand cDNA was synthesized from 200 ng total RNAs using Moloney murine leukemia virus (MMLV) reverse transcriptase (Promega Corporation, Madison, WI, USA) and oligo (dT) as a primer according to the manufacturer's instructions. All quantitative PCRs were performed in triplicates on a LightCycler 480 instrument using a LightCycler 480 SYBR green I master mix (Roche, Penzberg, Germany) according to the manufacturer's recommendations with gene-specific primers (S4 Table). Primers for *stylin-01* were designed in the non-mutated 5' end sequence, which is still present in the genomes of both mutant lines. Amplification efficiencies were analyzed with the LinRegPCR 512 software [29]. The genes encoding the mitochondrial malate dehydrogenase (Mdh2) and the NADH dehydrogenase [ubiquinone] iron-sulfur protein 8 (Ndufs8) exhibit stable expression across samples and tissues. *Mdh2* was used as reference gene. *Ndufs8* was used as internal control to compare the expression of different stylins within a single aphid line. Relative expressions were determined by using the $2^{-\Delta\Delta CT}$ method [30].

### Analysis of aphid feeding behavior

The Electrical Penetration Graph (EPG) method was used as described by [31] to investigate if specific stylet penetration activities known to be associated with acquisition and inoculation were altered in individuals of mutant lines. A Giga-8 DC-EPG device (EPG Systems, Wageningen, The Netherlands) was used to monitor probing and ingestion activities of N2 instar nymphs (for detailed description, see S1 Text). A total of 21–22 aphid nymphs per line were recorded, after they underwent a one-hour pre-acquisition starvation period before EPG recordings began on 21 dpi CaMV-infected plants. To assess potential mutation effects on feeding behavior during acquisition, recordings were conducted for 10 minutes from the start until interruption on CaMV-infected turnip plants. After this 'acquisition phase', the nymphs were individually transferred to young, healthy turnip plants for an additional 10 minutes of recording, constituting the 'inoculation phase'. Each recording was then semi-automatically labelled using the software A2-EPG [32], i.e., each recording was scanned by the software but then reviewed manually to check/correct/add labels. The following waveforms in particular were focused: waveforms C (complex of different patterns, correlated with stylet tips somewhere between the epidermis and phloem), potential drop (pd) waveforms (intracellular punctures of plant cells by the stylet tips), subphases II-3 and II-2 of the waveform pd associated with CaMV acquisition and inoculation, respectively [9]. We then calculated the following non-sequential variables for each individual: time from start of the probe to first pd (t > 1pd); duration of pd subphase II-2 (d_pdII-2); duration of pd subphase II-3 (d_pdII-3) [24]. From these individual values, we estimated the Waveform Duration per Insect [WDI, as defined by [33]] for each of the three aphid lines.

### Protein bioinformatics analysis

The three-dimensional protein structures were predicted via ColabFold v1.5.5 (AlphaFold2 using MMseqs2) [34]. We used an *ad hoc* pipeline to identify, in the Aphididae fully annotated genomes, available in GeneBank (NCBI) [35], putative orthologues of the five stylin genes of *A. pisum*: *stylin-01* (NM_001162314.1), *stylin-02* (NM_001162671.2), *stylin-03* (NM_001161959.2), *stylin-04* (NM_001172260.1) and *stylin-04bis* (NM_001172268.2). The protein sequences were used in a BLAST+ [36] search for sequences on GeneBank; the proteins retrieved were analyzed with Orthofinder [25] that returned for the sequences obtained from the 11 aphid species (see S6 Table for full list) three ortho-groups. The protein

sequences in the ortho-group of Stylin-01 and -02 were aligned with MAFFT [37,38], and the phylogenetic inference was realized using IQTree2 [39] with 1000 bootstraps and visualized using FigTree [http://tree.bio.ed.ac.uk/software/figtree/].

## Statistical analysis

Statistical analysis were performed using tests adapted to each data set (provided in S1 Data). Differences in aphid transmission efficiencies were tested using a binomial generalized linear model (GLM) followed by a post-hoc Tukey's honestly significant difference (HSD) test. Differences in stylets labeling were analyzed by GLM. The qPCR data were analyzed using a one-way analysis of variance (ANOVA), followed by the least significant difference (LSD) test. Differences in EPG variables and the duration of nymph development were analyzed using Kruskal-Wallis test. Different letters indicate significant differences among groups, and asterisks denoted statistical significance between two groups (*$P < 0.05$, **$P < 0.01$, and ***$P < 0.001$). All the data were analyzed using SPSS Statistics 20.0 software or R Stats Package [40], and plotted using GraphPad Prism version 9.

## Supporting information

**S1 Text. Supplementary Material and Methods.** Setup complementary information for the analysis of aphid feeding behavior.
(PDF)

**S1 Fig. Alignment of translated amino acid sequences of the Stylin-01 proteins encoded by the two alleles of Sty01-KO and Sty01-Cter aphid lines with the WT alleles.** Conserved amino acids are highlighted with a red background color. Predicted signal peptide sequences are underlined in black. The WT RR-1 chitin-binding domain is indicated with a double blue arrow. The multiple alignment of amino acid sequences was performed using Clustal2.1 (http://www.clustal.org/clustal2/). Alignment results were presented by ESPript 3.0 (https://espript.ibcp.fr/ESPript/ESPript/index.php). The signal peptides were predicted using the SignalP 6.0 server (https://services.healthtech.dtu.dk/services/SignalP-6.0/). The prediction and localization of chitin-binding domain were conducted in cuticleDB (http://bioinformatics2.biol.uoa.gr/cuticleDB/index.jsp).
(PDF)

**S2 Fig. Schematic representation of the antibodies used to detect the Stylin-01 or Stylin-02 proteins in wild-type and mutant aphid lines.** (A-C) Schematic representation of the domains of the mature Stylin-01 and Stylin-02 proteins as represented in Figure 1 with peptides targeted by the different antibodies indicated as black lines. (A) Peptides targeted by anti-1–09 or anti-1–11 antibodies initially produced in Webster et al., 2018 [14] to detect wild-type Stylin-01 protein. (B) Peptide targeted by anti-1-L14Cter specifically produced in this study to detect the mutated C-terminus of the protein encoded by the Sty01-Cter allele 2 of *stylin-01* gene. (C) Peptides of Stylin-02 targeted by anti-1–07 or anti-1–11 antibodies. (+) and (-) respectively indicate whether one protein should be detected or not by the antibody, respectively. The N-terminal (Nter) and C-terminal (Cter) domains surrounding the RR-1 chitin-binding-domain are indicated in green, red and blue respectively. Mutations in the amino acid sequences (Mut) are indicated in purple.
(PDF)

**S3 Fig. Duration of nymphs' development in wild-type and mutant lines.** Nymphal development of WT, Sty01-KO and Sty01-Cter aphid lines at 18°C under 16:8 h (day:night) on faba beans. The diagram results from two independent biological replicates and at least 20 aphid nymphs examined per aphid line. The sampling time performed at late N4 stage 46 h post-molt is indicated by a red arrow.
(PDF)

**S4 Fig. Detection of Stylin-02 protein in whole-body crude extracts of wild-type and mutant aphids by western blot analysis.** Each vertical lane corresponds to the protein crude extracts of 10 aphid whole bodies. Whole-body crude extracts proteins from the WT (lane 1), Sty01-KO (lane 2), and Sty01-Cter (lane 3) were analyzed by SDS-PAGE and stained with Coomassie blue to compare the protein amount loaded on the gel, or transferred onto a nitrocellulose membrane prior to immunolabeling with anti-1–07 antibody targeting Stylin-02 (S2C Fig and S1 Table).
(PDF)

**S5 Fig. Phylogenetic analysis of RR-1 stylins and species consensus tree.** (A) Phylogenetic tree generated with IQtree based on a MAFFT alignment and edited in FigTree. *Acyrthosiphon pisum* Stylin-01 and Stylin-02 are indicated by a red arrow. Green backgroup for the cluster containing Stylin-01 and yellow background for the cluster containing Stylin-02. (B) The species consensus tree is based on Jousselin et al., 2024 [26] and Hardy et al., 2022 [27] of the Aphididae species for which we retrieved orthologs.
(PDF)

**S1 Table. Antibodies used in the study.**
(PDF)

**S2 Table. Duration of the development of wild-type and mutant nymphs before adult molt at 18 °C and a 16/8 h (day/night) photoperiod.** Results are expressed as mean hours (± SE) of at least 20 nymphs per aphid line.
(PDF)

**S3 Table. Detection of Stylin peptides at the surface of adult maxillary stylets in wild-type and mutant aphid lines.** Ac: acrostyle; Ed: edge at the apex of maxillary stylets. The fluorescence was scored "Strong labeling" for a strong signal evenly detected on the acrostyle surface; "Weak labeling" for a weak signal detected as dots or partially distributed on the acrostyle surface. Total over two independent biological replicates.
(PDF)

**S4 Table. List of the oligonucleotides used for qRT-PCR analyses.**
(PDF)

**S5 Table. Comparison of the feeding behavior of N2 nymphs of WT and mutant *A. pisum* lines.** Feeding behavior recorded for 10 minutes on turnip plants infected with CaMV (21 dpi) for EPG variables related to virus acquisition, and for 10 minutes on young healthy turnip plants for EPG variables related to virus inoculation. Time is expressed in seconds (mean values ± SE).
(PDF)

**S6 Table. Stylins orthologs.** Orthologs of *A. pisum* stylin proteins identified with Orthofinder, the orthogroup 1 were used in the IQTree analysis.
(XLSX)

**S1 Data. Supplementary Excel File including raw data.**
(XLSX)

## Acknowledgments

All rearing and experiments on insects have been performed on the Baillarguet Vectoplante platform, member of the Vectopole Sud Network (http://www.vectopole-sud.fr/).

## Author contributions

**Conceptualization:** Yu Fu, Nicolas Sauvion, Gaël Thébaud, Gaël Le Trionnaire, Stefano Colella, Marilyne Uzest.

**Formal analysis:** Yu Fu, Maëlle Deshoux, Elian Strozyk, Emmanuelle Jousselin, Nicolas Sauvion, Gaël Thébaud, Stefano Colella, Marilyne Uzest.

**Funding acquisition:** Emmanuelle Jousselin, Stefano Colella, Marilyne Uzest.

**Investigation:** Yu Fu, Maëlle Deshoux, Bastien Cayrol, Sophie Le Blaye, Emma Achard, Sylvie Hudaverdian, Elodie Pichon, Elian Strozyk, Marilyne Uzest.

**Project administration:** Stefano Colella, Marilyne Uzest.

**Resources:** Sylvie Hudaverdian, Romuald Cloteau, Nathalie Prunier-Leterme, Gaël Le Trionnaire.

**Supervision:** Stefano Colella, Marilyne Uzest.

**Visualization:** Yu Fu, Bastien Cayrol, Stefano Colella.

**Writing – original draft:** Yu Fu, Stefano Colella, Marilyne Uzest.

**Writing – review & editing:** Maëlle Deshoux, Emmanuelle Jousselin, Nicolas Sauvion, Gaël Thébaud, Gaël Le Trionnaire, Stefano Colella, Marilyne Uzest.

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
