## [Decision Letter · Decision Letter 0]

8 Apr 2025

PPATHOGENS-D-25-00024

Stylet cuticular gene-directed mutagenesis impairs the pea aphid vector capacity to transmit a plant virus

PLOS Pathogens

Dear Dr. Uzest,

Thank you for submitting your manuscript to PLOS Pathogens. After careful consideration, we feel that it has merit but does not fully meet PLOS Pathogens's publication criteria as it currently stands. Therefore, we invite you to submit a revised version of the manuscript that addresses the points raised during the review process.

Please submit your revised manuscript within 30 days Jun 07 2025 11:59PM. If you will need more time than this to complete your revisions, please reply to this message or contact the journal office at plospathogens@plos.org. Please include the following items when submitting your revised manuscript:

We look forward to receiving your revised manuscript.

Kind regards,

Christophe Ritzenthaler

Academic Editor

PLOS Pathogens

Savithramma Dinesh-Kumar

Section Editor

PLOS Pathogens

Sumita Bhaduri-McIntosh

Editor-in-Chief

PLOS Pathogens

orcid.org/0000-0003-2946-9497

Michael Malim

Editor-in-Chief

PLOS Pathogens

orcid.org/0000-0002-7699-2064

**Journal Requirements:**

- ® on pages: 24 and 27.

Potential Copyright Issues:

- Please confirm (a) that you are the photographer of Figure 1A, and 5, or (b) provide written permission from the photographer to publish the photo(s) under our CC BY 4.0 license.

6) Please amend your detailed Financial Disclosure statement. This is published with the article. It must therefore be completed in full sentences and contain the exact wording you wish to be published. Please ensure that the funders and grant numbers match between the Financial Disclosure field and the Funding Information tab in your submission form. Note that the funders must be provided in the same order in both places as well. State the initials, alongside each funding source, of each author to receive each grant. For example: "This work was supported by the National Institutes of Health (####### to AM; ###### to CJ) and the National Science Foundation (###### to AM).".

**Reviewers' Comments:**

Reviewer's Responses to Questions

**Part I - Summary**

Reviewer #1: The manuscript describes an extensive analysis of Cas9 mutants of the stylet proteins which participate in CaMV virus transmission. The results are internally consistent and appropriate.

The novelty of this work is repeatedly noted – though the generation of the mutants was previously published.

The authors have done a good job of making quantitative presentation of qualitative observations.

Reviewer #2: Aphids are serious agricultural pests, notorious for transmitting hundreds of plant viruses, including Cauliflower mosaic virus (CaMV). CaMV transmission by aphids relies on a helper mechanism involving the viral non-structural protein P2 and its receptors on the surface of the acrostyle, a specialized structure at the tip of the insect’s stylet. Stylin proteins are prime candidates for these receptors. In their study, Fu et al. developed two mutant aphid lines with edits to the Stylin-01 gene: one is a complete knockout, and the other produces a modified version of the protein with changes to its exposed C-terminal domain. Using these mutants, the researchers showed that fully functional Stylin-01 is essential for efficient CaMV transmission by aphids. This research significantly enhances our understanding of the critical role this cuticular component plays in plant-virus-vector interactions. This manuscript is well-written and only a few minor modifications and discussion are required.

Reviewer #3: The long-standing question of the biochemical basis of how plant viruses are transmitted by insects to the host plant is key to the development of new, environmentally approaches to controlling their transmission. These viruses can replicate within the insect host while others have a direct transmission route from infected stylets or other mouthparts.

Previous studies had used RNAi approaches to down-regulate target genes however these are, by nature, transient, variable and somewhat inconclusive. Here the authors utilized two previously generated CRISPR/Cas9 mutants in a stylet gene to determine the role of this gene in virus transmission. One was essentially a completely dysfunctional mutation whole the second had retained most of the amino-terminal sequence but has lost the carboxy-terminal domain.

The experiments were well conducted and clearly show the role that the stylin-1 gene plays in transmission of the virus. They implicate the ability of the acrostyle to successfully bind the P2 protein of the virus and also show up-regulation effects on another stylin in these mutants.

I recommend the manuscript be accepted. It is short and concise, but will be of significant interest in the field.

**Part II – Major Issues: Key Experiments Required for Acceptance**

Reviewer #1: The raw data for the figures should be made available via a data repository, or provided as a supplemental spreadsheet / document. This is particularly true given the nature of statistical packages used for the analysis.

Reviewer #2: 1. In lines 119-121, the authors mention that the Sty01-Cter line lacks the 11 C-terminal amino acids that form the surface-exposed domain of the acrostyle, which is thought to interact with viruses. Did the researchers test whether this domain is essential for binding to CaMV? To properly verify this interaction, co-immunoprecipitation (Co-IP) or immunofluorescence co-localization experiments should have been conducted.

2. Fig 3 demonstrates a reduction in the number of positively labeled stylets for both mutant lines compared to WT aphids. However, it appears that a significant majority of the mutant aphids (over 80%) still retain the ability to bind CaMV P2 proteins and thus can transmit the virus. This raises the question of whether Stylin-01 is indeed the most critical protein regulating viral transmission. While the gene knockout has a statistically significant effect, the impact on virus transmission does not seem substantial enough to conclusively establish Stylin-01 as the primary regulator. This observation suggests that other factors or proteins might also play crucial roles in CaMV transmission, which deserves discussion.

Reviewer #3: (No Response)

**Part III – Minor Issues: Editorial and Data Presentation Modifications**

Reviewer #1: The writing style is at times cumbersome with excessive use of comas and prepositions. The PDF is provided with several dozen suggested edits.

The presentation of statistical includes non-parametric tests associated with non-normal distribution. Additional detail, including the assessment of normality would seem appropriate, even though the work understandably has limited numbers (making normality tests problematic). This statistical analysis seems to rely on the authors use of packages, more so than an understanding of the underlying statistical tests.

The discussion is quite redundant with the results, including reiterated references to the figures rather than extending the analysis to a broader perspective. This section could be considerably shortened.

The effort made in high quality presentation of graphics and tables (including supplemental) is noted. The overall quality and quantity of research represents a excellent addition to the literature.

Reviewer #2: 1. The antibodies used for western blotting and immunofluorescent localization, particularly those against Sty01-Cter and Sty01-KO in Fig 1, 3, and 5, should be clearly indicated either in the figures themselves or thoroughly described in the figure legends. This will help readers better understand and interpret the data presented in the figures.

2. Fig 3B presents the number of stylets observed for each group from a single experiment batch. To establish the reproducibility of these results, it is a standard practice to conduct at least 3 independent experimental batches. Without multiple batches, the results may lack statistical significance and robustness. It is crucial to know whether the authors conducted different experimental batches to validate their findings.

Reviewer #3: (No Response)

PLOS authors have the option to publish the peer review history of their article (what does this mean? ). If published, this will include your full peer review and any attached files.

**Do you want your identity to be public for this peer review?** For information about this choice, including consent withdrawal, please see our Privacy Policy .

Reviewer #1: **Yes: ** Wayne R. Curtis

Reviewer #2: No

Reviewer #3: No

**Figure resubmission:**
---

## [Editor Report · Decision Letter 1]

7 May 2025

Dear Dr Uzest,

We are pleased to inform you that your manuscript 'Stylet cuticular gene-directed mutagenesis impairs the pea aphid vector capacity to transmit a plant virus' has been provisionally accepted for publication in PLOS Pathogens.

Best regards,

Christophe Ritzenthaler

Academic Editor

PLOS Pathogens

Savithramma Dinesh-Kumar

Section Editor

PLOS Pathogens

Sumita Bhaduri-McIntosh

Editor-in-Chief

PLOS Pathogens

orcid.org/0000-0003-2946-9497

Michael Malim

Editor-in-Chief

PLOS Pathogens

orcid.org/0000-0002-7699-2064
---

## [Editor Report · Acceptance letter]

Dear Dr Uzest,

We are delighted to inform you that your manuscript, "Stylet cuticular gene-directed mutagenesis impairs the pea aphid vector capacity to transmit a plant virus," has been formally accepted for publication in PLOS Pathogens.

Best regards,

Sumita Bhaduri-McIntosh

Editor-in-Chief

PLOS Pathogens

orcid.org/0000-0003-2946-9497

Michael Malim

Editor-in-Chief

PLOS Pathogens

orcid.org/0000-0002-7699-2064